# Predictors of Maternal Serum Concentrations for Selected Persistent Organic Pollutants (POPs) in Pregnant Women and Associations with Birth Outcomes: A Cross-Sectional Study from Southern Malawi

**DOI:** 10.3390/ijerph20075289

**Published:** 2023-03-28

**Authors:** Mphatso Mwapasa, Sandra Huber, Bertha Magreta Chakhame, Alfred Maluwa, Maria Lisa Odland, Victor Ndhlovu, Halina Röllin, Shanshan Xu, Jon Øyvind Odland

**Affiliations:** 1Department of Public Health and Nursing, Norwegian University of Science and Technology, 7491 Trondheim, Norway; 2Department of Laboratory Medicine, University Hospital of North Norway, 9038 Tromsø, Norway; 3School of Maternal, Neonatal and Reproductive Health, Kamuzu University of Health Sciences, Blantyre 312225, Malawi; 4Directorate of Research and Outreach, Malawi University of Science and Technology, Thyolo 310106, Malawi; 5Malawi Liverpool Wellcome Trust Clinical Research Programme, Blantyre 312233, Malawi; 6Department of Obstetrics and Gynaecology, St. Olav’s Hospital, 7030 Trondheim, Norway; 7Institute of Life Course and Medical Sciences, University of Liverpool, Liverpool L7 8TX, UK; 8School of Life Sciences and Health Professions, Kamuzu University of Health Sciences, Blantyre 312225, Malawi; 9School of Health Systems and Public Health, Faculty of Health Sciences, University of Pretoria, Pretoria 0002, South Africa; 10Centre for International Health, Department of Global Public Health and Primary Care, University of Bergen, 5009 Bergen, Norway

**Keywords:** persistent organic pollutants, birth outcomes, Southern Malawi

## Abstract

Population exposure to persistent organic pollutants (POPs) may result in detrimental health effects, especially to pregnant women, developing foetuses and young children. We are reporting the findings of a cross-sectional study of 605 mothers in their late pregnancy, recruited between August 2020 and July 2021 in southern Malawi, and their offspring. The aim was to measure the concentrations of selected POPs in their maternal serum and indicate associations with social demographic characteristics and birth outcomes. A high level of education was the main predictor of *p,p*′-DDE (*p* = 0.008), *p,p*′-DDT (*p* < 0.001), cis-NC (*p* = 0.014), *o,p*′-DDT (*p* = 0.019) and *o,p*′-DDE (*p* = 0.019) concentrations in maternal serum. Multiparity was negatively associated with *o,p*′-DDE (*p* = 0.021) concentrations. Maternal age was also positively associated (*p,p*′-DDE (*p* = 0.013), *o,p*′-DDT (*p* = 0.017) and *o,p*′-DDE (*p* = 0.045) concentrations. Living in rural areas was inversely associated with high maternal serum concentrations of *p,p*′-DDT (*p* < 0.001). Gestational age was positively associated with *p,p*′-DDE (*p* = 0.031), *p,p*′-DDT (*p* = 0.010) and *o,p*′-DDT (*p* = 0.022) concentrations. Lastly, an inverse association was observed between head circumference and t-NC (*p* = 0.044), Oxychlordane (*p* = 0.01) and cis-NC (*p* = 0.048). These results highlight the need to continue monitoring levels of POPs among vulnerable populations in the southern hemisphere.

## 1. Introduction

Persistent organic pollutants (POPs) are carbon-based, man-made synthetic chemical substances produced mainly for commercial purposes, such as for pests and disease control, crop production and industrial use [1]. These compounds have very long half-lives when released into the environment. They accumulate in the environment and adversely affect the food chain and living organisms, including humans, because of their lipid solubility and resistance to biodegradation [2]. However, it was not until 1962, when Rachel Carson, through her book entitled *Silent Spring*, revealed that these substances could be both uncontrollable and unexpectedly toxic [3]. Globally, there is growing evidence suggesting the link between these compounds and the occurrence of various noncancerous and cancerous health outcomes. Among many adverse health effects, there is growing concern about the possible association between serum POPs and reproductive health, wheezing in infants or children, Type 2 diabetes mellitus (DMT2), liver cancer, breast cancer and more recently, Parkinson’s disease [4,5,6,7].

Although the use and production of most POPs were restricted following the Stockholm Convention in 2001, some are still present in different environments and in living organisms, including humans, due to their bio-accumulative and long-persistent natures [8]. Twelve key compounds that were included in the Stockholm Convention were aldrin, chlordane, dichlorodiphenyl trichloroethane (DDT), dieldrin, endrin, heptachlor, hexachlorobenzene, mirex, toxaphene, polychlorinated biphenyls (PCBs), polychlorinated dibenzo-*p*-dioxins (dioxins) and polychlorinated dibenzofurans (furans) [9].

Since Malawi´s economy is largely dependent on agriculture, with about 80% of the population employed in this sector [10], there is an increasing demand for the use of pesticides. It is estimated that, in general, Malawi uses at least 2000 metric tons of pesticides annually, of which 70% are used for agriculture [11]. While there is extensive use of pesticides for agriculture in Malawi, there are limited data on the determinants and concentrations of POPs in the maternal serum. Various research studies have established that POPs are permeable through the placenta, and the findings have suggested that they can impair the growth and development of the foetus [12,13,14] and that exposure to POPs in the early stages of life (in utero) may pose a critical risk to health.

Most of the monitoring and research on POPs has been carried out in countries situated in the northern hemisphere. On the other hand, in the southern hemisphere, there has been limited research on predictors for POP concentrations in maternal serum and their associations with reproductive health outcomes. Research on POPs and human health outcomes, particularly concerning reproductive health in Malawi and other countries in the southern hemisphere, is of importance for global health [15]. This study, therefore, aimed to examine predictors of the serum concentrations of different POPs in delivering women and investigate the relationship of the POP concentrations with birth outcomes in Malawi.

## 2. Materials and Methods

### 2.1. Study Design, Population and Sites

Details on the study design, population and sites were described in detail in the previous publication of our study [16]. In brief, this cross-sectional study of pregnant women and their offspring was conducted in the southern region of Malawi, and study participants were recruited between August 2020 and July 2021.

The inclusion criteria included all pregnant women who were in the late stages of their pregnancy; 16 years of age and above; permanent residents of the Blantyre, Chiradzulu and Thyolo districts; and willing to voluntarily sign an informed consent statement. All women with serious medical conditions present at the time of enrolment, high risk pregnancies with prognoses to be referred to a tertiary health care level, or were also participating in another study were excluded. In this regard, a total of 771 pregnant women were assessed for eligibility and 605 were recruited for the study. However, only 564 women–neonate pairs were included in the final study population for statistical analysis on the associations between POP concentrations, as 41 either declined to have their biological sample collected, or the volume of the sample was insufficient. Figure 1 describes the details of the number of participants assessed for inclusion and recruited.

### 2.2. Study Questionnaire

The questionnaire that was used for the collection of information about mothers and neonates was already described in the first publication of our study [16]. Briefly, pregnant women´s personal characteristics, reproductive history, socioeconomic status, lifestyle, environmental characteristics, diet and neonatal data were collected just before delivery using a provider-administered questionnaire.

### 2.3. Sample Collection and Preliminary Analysis

Methods for the whole blood sample collection and preliminary analysis used in the present study were adapted from the international quality control system QA/QC established by the Centre de Toxicologie du Quebec [17]. The detailed process of the sample collection and preliminary analysis were described in our first publication for the study [16].

### 2.4. Serum Sample Analysis

Serum samples were analysed at the Environmental Pollutant Laboratory at the University Hospital of North Norway according to the previously published method by Huber et al. [18]. In short, a Tecan Freedom Evo 200 (Männedorf, Switzerland) liquid-handling workstation was applied for automated sample preparation. First, 150 µL aliquots of serum samples were diluted prior to extraction on reversed phase 96-well plates, which was followed by clean-up on normal phase extraction columns. Gas chromatography atmospheric pressure ionisation coupled to tandem mass spectrometers (GC-API-MS/MS; Waters, Milford, MA, USA) were used for instrumental analysis. The API was conducted in positive mode under charge transfer conditions. For detection on the mass spectrometer, the multiple reaction monitoring mode was applied with two specific transitions for the individual analytes. Quantification was performed using the Masslynx and Targetlynx software (Version 4.1 and 4.2, Waters, Milford, MA, USA) and achieved by the internal-standard method with isotope-labelled compounds.

### 2.5. Measurement of Birth Outcomes

The following birth outcome variables were measured soon after delivery: gestational age (weeks), birth weight (kg), birth length (cm) and head circumference (cm) [16]. Gestational age was estimated at delivery by referring to the antenatal health records and the ponderal index was used to estimate the nutritional status of the newborns and calculated using the following formula [16,19]:(1)$$ponderal\;index=\frac{weight\;(kg)}{{height}^{3}\;(m^{3})}$$

### 2.6. Statistical Analysis

Data analysis was performed using the statistical software Stata for Mac (SE standard version 17; College Station, TX, USA). Our analysis focused on compounds with the detection frequency of ≥50%. With this criterion, the compounds that qualified for the analyses were: HCB, *p,p*′-DDE, t-NC, *p,p*′-DDT, Oxychlordane, cis-NC, *o,p*′-DDT and *o,p*′-DDT (Table 1). To control for missing data, concentrations of selected compounds below the limit of detection (LOD) were replaced by LOD/2. LODs were automatically calculated by the Masslynx software for each individual analyte and sample. Detailed information on the LODs is given in Appendix A. Descriptive statistics including arithmetic means, standard deviation (SD), median and minimum and maximum or proportion (%) were computed from the data set (Table 1). A significance level of *p* < 0.05 (two-tailed) was set for all analyses. The data were fitted into a multivariable regression model with maternal characteristics as the response variable and the significant POP concentrations as predictors. All POP concentrations were log-transformed before inclusion in the multivariable regression models due to non-normal distribution of the concentrations among the participants.

## 3. Results

### 3.1. Maternal Socio-Demographic and Neonates’ Anthropogenic Data

The socio-demographic and neonates’ anthropogenic data are described in our previously published article from this Malawi study [16]. The age of women recruited in the study ranged from 16 to 45 years, with a mean (SD) age of 24.8 (6.2) years. Just over half of the women (308) were recruited from the urban setting, and 297 were from rural areas. Women recruited from the urban setting were almost two years older than their counterparts recruited from rural areas. In this regard, the mean ages (SD) for pregnant women recruited from urban and rural areas were 25.6 (6.7) and 23.9 (5.7) years, respectively.

The data showed differences in gravidity, parity and educational levels between the urban and rural pregnant women. On this, nulliparity was significantly high in rural areas. In contrast, Para 1 and multiparity were statistically high among women recruited from urban areas, as compared to their counterparts from rural sites.

A small (38.6%) proportion of women recruited from urban areas either did not attend any formal school or were educated just up to the primary level. However, the majority (61.4%) attained an education up to the secondary or tertiary level. In contrast, the majority of pregnant women from rural areas did not attain higher educational levels (69.7% attained up to the primary level versus only 30.3% attaining the secondary/tertiary level). The use of tap water as the source of drinking water was very common (96.8%) among urban study participants. As for the rural setting, shallow wells and boreholes were collectively the common source of drinking water, representing 44.1% and 45.5%, respectively.

In total, 572 neonates (51.8% boys) were recruited in the present study. The mean birth weight, length and head circumference for the neonates recruited were 3.09 kg, 45.28 cm and 33.15 cm, respectively, in the overall sample. Detailed information about maternal socio-demographics and the neonates´ anthropogenic data according to their area of residence (urban versus rural) is also outlined in the Appendix A.

### 3.2. Maternal POP Serum Concentrations

Eight out of the 19 POPs with detection frequencies (DF) ≥50% were included in the statistical analysis. The eight compounds included in the statistical analysis were Hexachlorobenzene (HCB), 1-chloro-4-[2,2-dichloro-1-(4-chlorophenyl)ethenyl]benzene (*p,p*′-DDE), trans-Chlordane (t-NC), 1-chloro-4-[2,2,2-trichloro-1-(4-chlorophenyl)ethyl]benzene *(p,p*′-DDT), Oxychlordane, cis-Nonachlor (cis-NC), 1-chloro-2-[2,2,2-trichloro-1-(4-chlorophenyl)ethyl]benzene (*o,p*′-DDT) and 1-chloro-2-[2,2-dichloro-1-(4-chlorophenyl)ethenyl]benzene (*o,p*′-DDE). Table 1 provides descriptive statistics for the selected eight POPs in pg/mL (as wet weight concentration of the POP/unit serum). The two POPs with the highest median concentrations observed in this study were 478 pg/mL and 102 pg/mL for *p,p*′-DDE and HCB with detection frequencies of 99.3% and 99.8%, respectively. On the other end, the lowest median concentration assessed was 0.34 pg/mL for *o,p*′-DDE. Detailed information of the POPs that were included in the in the statistical analysis is presented in Table 1.

### 3.3. Concentrations of POPs in Maternal Serum and Associations with Maternal Characteristics

The POP concentrations followed an approximately normal distribution after logarithmic transformation. Maternal characteristics, namely age and parity, were automatically considered to be included in the multivariate analysis, as they are already known to be associated with POP concentrations in maternal serum based on previous studies. Furthermore, univariable linear regression analysis was used to determine the covariates that needed to be included in the multivariable linear regression models. In this regard, the following factors were selected to be included in the multivariable analysis: maternal educational level, previous breast-feeding and beef and goat meat consumption frequencies. Table 2 and Table 3 show the detailed results of the univariate and multivariable analyses, respectively, between maternal characteristics and different POP concentrations in maternal serum. 

Multivariable linear regression models of the aforementioned variables provided an overall description of the main factors influencing the concentrations of the selected POPs. Comprehensive descriptions of the main maternal risk factors related to the serum POP levels are given in Table 2 and Table 3.

After adjusting for maternal age, parity, area of residence (urban versus rural), source of drinking water and beef and goat meat consumption, participants educated up to either secondary or tertiary levels had significantly higher concentrations of *p,p*′-DDE (*p* = 0.008), *p,p*′-DDT (*p* < 0.001), cis-NC (*p* = 0.014), *o,p*′-DDT (*p* = 0.007) and *o,p*′-DDE (*p* = 0.019) than those who were only educated up to the primary level. Similarly, greater maternal ages were statistically associated with higher maternal serum concentrations of *p,p*′-DDT (*p* = 0.013), *o,p*′-DDT (*p* = 0.017) and *o,p*′-DDE (*p* = 0.045).

Multiparity was significantly associated with decreased maternal serum concentrations of some POPs. This was observed for *o,p*′-DDT (*β* = −0.927; 95% CI: −1.713 to −0.141; *p* = 0.021). Living in rural areas was inversely associated with maternal serum concentrations of *p,p*′-DDT (*β* = −1.302; 95% CI: −1.871 to −0.733; *p* < 0.001).

Increased maternal t-NC serum concentrations were positively associated with the use of tap lake/shallow well (*β* = 0.929; 95% CI: 0.430 to 1.427; *p* < 0.001) and borehole water (*β* = 0.742; 95% CI: 0.260 to 1.224; *p* = 0.003) as sources of drinking water, as opposed to the use of tap water. Similarly, high levels cis-NC were also positively associated with the use of lake/shallow well (*β* = 1.311; 95% CI: 0.642 to 1.980; *p* < 0.001) and borehole (*β* = 1.026; 95% CI: 0.379 to 1.672; *p* = 0.002) water. Lastly, women who reported previous breast-feeding had statistically lower *p,p*′-DDE serum concentrations (*β* = −0.643; 95% CI: −1.174 to −0.113; *p* = 0.018) in reference to those who without such history.

### 3.4. Maternal Dietary Habits and Level of Education

Human exposure to POPs is mainly through the ingestion of foods of animal origin, such as meat, fish, dairy items and eggs [20]. In this regard, data on diet and other maternal characteristics, such as the maternal educational level, were also explored. Univariate analysis results showed that the consumption of beef more than twice a week was statistically significantly associated with higher maternal serum concentrations of HCB (*β* = 0.900; 95% CI: 0.035 to 0.145; *p* < 0.001), *p,p*′-DDT (*β* = 0.544; 95% CI: 0.247 to 0.841; *p* < 0.001) and *o,p*′-DDE (*β* = 0.209; 95% CI: 0.038 to 0.379; *p* = 0.017). A similar trend of associations was also observed for the consumption of goat meat as follows: HCB (*β* = 0.076; 95% CI: 0.024 to 0.129; *p* = 0.004), *p,p*′-DDT (*β* = 0.351; 95% CI: 0.066 to 0.636; *p* = 0.016) and *o,p*′-DDE (*β* = 0.206; 95% CI: 0047 to 0.369; *p* = 0.013). However, these associations were not observed in the multivariable linear regression model.

### 3.5. POP Concentrations in Maternal Serum and Their Associations with Birth Outcomes

In order to investigate the possible associations between the selected POPs and birth outcomes, multivariable linear regression models were applied and adjusted for maternal age, area of residence (urban versus rural), maternal educational level, parity and source of drinking water. As shown in Table 4, multivariable linear regression analysis showed positive associations between the natural log-transformed maternal serum concentrations of *p,p*′-DDE (*β* = 0.087; 95% CI: 0.008 to 0.166; *p* = 0.031), *p,p*′-DDT (*β* = 0.110; 95% CI: 0.193 to 0.166; *p* = 0.01), *o,p*′-DDT (*β* = 0. 0.115; 95% CI: 0.016 to 0.213); *p* = 0.022) and gestational age. Furthermore, statistically significant inverse associations were observed between natural log-transformed maternal serum concentrations of t-NC (*β* = −0.053; 95% CI: −0.105 to −0.0015; *p* = 0.044), Oxychlordane (*β* = −0.071; 95% CI: −0.123 to −0.017; *p* = 0.010), cis-NC (*β* = −0.070; 95% CI: −0.140 to −0.006; *p* = 0.048) and head circumference of the neonates. No significant associations were observed between the selected POPs and birth length, birth weight or ponderal index (*p* > 0.05).

## 4. Discussion

Maternal level of education, age and parity were the main predictors of concentrations of POPs in the maternal serum. In this regard, increased maternal serum POP concentrations were found to be associated with increased levels of maternal education, age and parity. Living in urban settings was associated with increased maternal serum concentrations of HCB and *p,p*′-DDT. Some POPs (*p,p*′-DDE, *p,p*′-DDT and *o,p*′-DDT) were negatively associated with gestational age. Furthermore, t-NC, cis-NC and Oxychlordane were inversely associated with head circumference. Notably, *o,p*′-DDT was positively associated with gestational age.

The maternal serum POP concentrations observed in this study were generally low, as compared to other published studies conducted elsewhere [15,21,22] but comparable to the study conducted by Steinholt et al. in Cambodia [19]. In this regard, the median (min–max) maternal serum *p,p*′-DDT concentrations observed in these two studies were almost consistent (35.5 (0.43–1700) pg/mL ww for the current study and 33.0 (3–519) pg/mL ww for the Cambodian study). It was also noted that the median HCB concentration in the maternal serum observed in the present study was high (102 (0.57–203) pg/mL) compared to the Cambodian study (44.0 (10–196) pg/mL). In contrast, the median (min–max) concentrations of *p,p*′-DDE and *o,p*′-DDT observed in the present study were on the lower side (448 (0.10–23,600) vs. 720.0 (20–8047) pg/mL, and 5.10 (0.11–163) vs. 20.0 (20–20) pg/mL, respectively) and *o,p*′-DDE was on the higher side (8.00 (8–10) vs. 0.34 (0.04–8.9) pg/mL) compared to the Cambodian study.

For DDT and its metabolites, *p,p*′-DDE was the most abundant compound detected in the maternal serum. These findings are similar to previous studies from Cambodia and Vietnam [19,23]. This is because *p,p*′-DDE is the most common and stable metabolite of *p,p*′-DDT; hence, it is the most likely to be detected in both environmental and human samples. Furthermore, various studies worldwide have shown that the detection frequency of *p,p*′-DDE is 100% [24,25,26] despite restrictions on its production and use over 40 years ago. In addition, our dataset showed no indication of a fresh DDT-exposure source, since the concentrations of *p,p*′-DDT were rather low, as compared to *p,p*′-DDE.

The level of education was the main predictor of POP concentrations in the maternal serum. Generally, high educational levels were significantly associated with increased maternal serum concentrations of most of the POPs. This was the case with *p,p*′-DDE (*p* = 0.008), *p,p*′-DDT (*p* < 0.001), cis-NC (*p* = 0.014), *o,p*′-DDT (*p* = 0.007) and *o,p*′-DDE (*p* = 0.019). The positive statistically significant associations between the compounds *p,p*′-DDE and *p,p*′-DDT and educational level were consistent with the results found by Steinholt et al. in another study conducted in Cambodia [19]. This could be explained by differences in diets between the two groups. Human exposure to POPs is mainly through ingestion of foods of animal origin, such as meat, fish, dairy items and eggs [20]. Although the univariate analysis showed statistically significant positive associations between consumption of beef and goat and the concentrations a number of POPs (HBC, *p,p*′-DDT and *o,p*′-DDE), similar associations were not observed in the multivariable linear regression analysis.

Just as with maternal educational level, higher maternal age was also statistically significantly associated with higher maternal serum concentrations of *p,p*′-DDT, *o,p*′-DDT and *o,p*′-DDE. Concentrations of the above POPs in their serum were higher in older women than in younger women. This was expected, as high levels of most persistent organic compounds in the maternal serum increase with age, as previously reported in published literature [23,27,28,29,30]. It was also observed that living in urban areas was significantly associated with high concentrations of *p,p*′-DDT in the maternal serum. This could be attributed to different lifestyles, especially related to diet. For instance, pregnant women living in urban setting are more likely to consume meat-based foods frequently, as compared to rural residents. Furthermore, women residing in urban areas have high chances of consuming food that is contaminated with DDT and its metabolites, as most food stuffs in supermarkets are imported from South Africa, where DDT is still used for malaria prevention. However, there is a need for a large study on the effect of diet on the concentrations of maternal POPs to ascertain the above hypotheses.

Multiparity was associated with decreased maternal serum concentrations of *o,p*′-DDT. These results are similar to other studies that have also revealed that parity is negatively associated with maternal serum concentrations of some POPs [23,30] due to the fact that these compounds are permeable through the placenta and are transferred to the growing foetus [18,29,31,32,33]. Thus, a negative association between *p,p*′-DDT and parity is attributed to mother-to-foetus transfer of these compounds through the placenta. Furthermore, a large proportion of women (58.9%) recruited in this study reported previous breast-feeding. Since POPs are lipophilic in nature, they tend to be secreted into breast milk [34,35,36,37,38], and the inverse association observed can also be explained by depletion of these compounds in the maternal serum due to the transfer from mother to the baby through breast-feeding.

In the present study, positive associations were observed between gestational age and each of the following POPs: *p,p*′-DDE, *p,p*′-DDT and *o,p*′-DDT. These findings concur with results from a study by Jusko et al., in which a positive association was observed between *p,p*′-DDT and *o,p*′-DDT with gestational age [39]. However, the findings in the present study are in contrast to the results in a study by B J Wojtyniac et al. in the Kharkiv cohort, which found that lower gestational ages were associated with an increase in log *p,p*′-DDE exposure [40]. The difference in results between these two studies could be due to the sample size, regional distribution, data analysis methods and selection of confounding factors. In addition, we also observed a statistically significant inverse association between some POPs (t-NC, Oxychlordane and cis-NC) and the head circumferences of the neonates, as reported in other studies. There are limited published data on studies on the association between t-NC, Oxychlordane and cis-NC with head circumference. However, the previously listed substances are endocrine disruptors; hence, they are more likely to affect foetal growth, which may result in smaller head circumferences of the neonates.

The main limitation of this study is its cross-sectional design that prevents causality associations and, therefore, causality associations cannot be shown. However, to our knowledge, this is the first study to explore concentrations of selected POPs in the maternal serum of pregnant women in Malawi and their associated pregnancy outcomes. The study revealed predictors of POP levels in maternal serum and their associations with birth outcomes. Another strength is that samples were drawn from different settings (urban and rural), which makes the results more representative of Malawian women.

## 5. Conclusions

Our study made a comprehensive effort to examine the determinants of selected persistent organic pollutant concentrations in maternal serum during pregnancy and their associations with social demographic characteristics and birth outcomes. This study gives a foundation upon which knowledge of the main predictors for POP levels in maternal serum among pregnant women at the delivery stage and from low–middle income African settings can be built. In this regard, maternal level of education, age and parity were the main predictors of the concentrations of POPs in the maternal serum. Although the concentrations of the POPs observed were generally low, there is still a need for more research to periodically monitor the concentrations of these compounds and the associated lifestyles during pregnancy. In addition, the results from the present study may be used to facilitate the identification of specific lifestyles that are associated with high concentrations of POPs in the maternal serum of women of child-bearing age.

## Figures and Tables

**Figure 1 ijerph-20-05289-f001:**
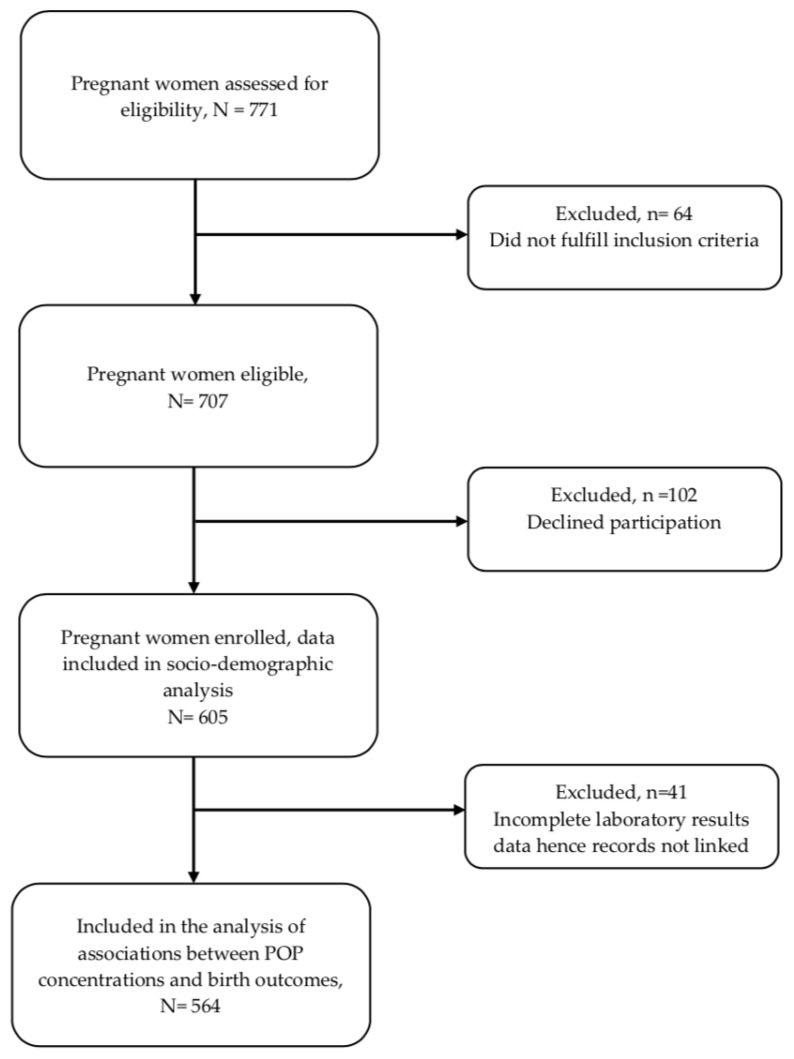
Flow chart of included and excluded participants.

**Table 1 ijerph-20-05289-t001:** Maternal serum concentrations (wet weight pg/mL) of POPs (*n* = 564).

POPs	% > DF	AM (SD) ^a^	GM (95% CI) ^b^	Median (Min–Max)
HCB	99.8	102 (20.7)	99.8 (97.4–104.19)	102 (0.57–203)
*p,p*′-DDE	99.3	878 (1575)	405 (359.5–456.7)	478 (0.10–23,600)
t-NC	98.1	73.7 (126)	35.3 (31.7–39.3)	34.5 (0.01–1240)
*p,p*′-DDT	86.9	81.3 (136)	30.9 (27.1–35.2)	35.5 (0.43–1700)
Oxychlordane	84.5	63.7 (102)	29.7 (26.6–33.1)	34.4 (0.05–1050)
cis-NC	84	13.1 (26.6)	4.3 (3.8–5.0)	5.4 (0.02–203)
*o,p*′-DDT	73.4	9.51 (16.0)	3.3 (2.8–3.8)	5.10 (0.11–163)
*o,p*′-DDE	58.4	0.55 (0.8)	0.34 (0.32–0.37)	0.34 (0.04–8.9)

**^a^** Arithmetic mean (AM) with standard mean deviation (SD). **^b^** Geometric mean (GM) with 95% CI. The detection frequency (DF) >50% of the samples. Missing data have been inputted as LOD/2.

**Table 2 ijerph-20-05289-t002:** Univariate linear regression of POP concentrations in blood serum and maternal characteristics.

Maternal Characteristics	HBC ^a^	*p*′*,p*′-DDE ^a^	t-NC ^a^	*p*′*,p*′-DDT ^a^	Oxychlordane ^a^	Cis-NC ^a^	*o*′*,p*′-DDT ^a^	*o*′*,p*′-DDE ^a^
**Maternal age (years)**	0.001	**0.06 ****	**−0.043 ****	**0.026 ***	**0.039 ****	**−0.037 ***	0.001	0.008
**Parity ^b^**								
Para 1	**0.068 ***	**−0.289**	**−0.353 ***	0.355	**−0.377 ***	−0.295	0.047	0.152
Multiparity	0.041	**−1.068 ****	**0.695 ****	0.276	**−0.724 ****	**−0.701 ****	−0.131	0.023
**Education of mothers ^c^**	**0.058 ***	**0.600 ****	0.154	**0.835 ****	0.136	0.273	**0.446 ***	**0.291 ****
**Area of residence ^d^** (urban vs. rural)	**−0.12 ****	−0.21	**0.343 ***	**−1.129 ****	**0.429 ****	**0.508 ***	−0.25	**−0.308 ****
**Source of drinking water ^e^**								
Lake/shallow well	**−0.12 ****	−0.113	**0.533 ****	**−1.174 ****	**0.486 ***	**0.790 ****	−0.28	**−0.248 ***
Borehole	**−0.080 ****	−0.27	**0.399 ***	**−0.816 ****	**0.433 ***	**0.561 ***	−0.139	**−0.391 ****
**Previous breast-feeding ^f^**	0.102	**−0.643 ****	**−0.575 ****	0.164	**−0.620 ****	**−0.568 ****	-0.035	0.044
**Beef consumption frequency ^g^**	**0.090 ***	0.111	−0.181	**0.544 ***	−0.231	−0.211	−0.03	**0.209 ***
**Egg consumption frequency ^h^**	0.003	0.115	0.018	**0.284**	−0.025	−0.017	0.022	**0.218 ***
**Fresh fish consumption frequency ^i^**	0.004	0.186	0.178	0.157	0.127	**0.313 ***	−0.027	**0.221 ***
**Dry fish consumption frequency ^j^**	0.024	−0.25	0.206	−0.341	−0.009	0.032	−0.342	−0.102
**Goat consumption frequency ^k^**	**0.076 ***	0.212	−0.124	**0.351 ***	−0.173	−0.165	0.041	**0.206 ***
**Green vegetables consumption frequency ^l^**	0.167	1.288	1.356	1.012	1.079	**2.439 ***	0.562	0.183

Values shown were simple linear regression analyses coefficients. ^a^ All POP concentrations were log 10 transformed before analysis. ^b^ Para 0 is used as a reference category. ^c^ None and primary educational level are the reference categories. ^d^ Urban is the reference category. ^e^ Tap water is the reference category. ^f^ No previous breast-feeding is the reference category. ^g–l^ Less than twice a week is used as the reference category. * *p* < 0.05; ** *p* < 0.001. Significant findings are printed in bold.

**Table 3 ijerph-20-05289-t003:** Multivariable linear regression of POP concentrations in blood serum and maternal characteristics.

Maternal Characteristics	HBC ^a^	*p*′*,p*′-DDE ^a^	t-NC ^a^	*p*′*,p*′-DDT ^a^	Oxychlordane ^a^	Cis-NC ^a^	*o*′*,p*′-DDT ^a^	*o*′*,p*′-DDE ^a^
**Maternal age (years)**	−0.003	−0.005	−0.017	0.037 *	−0.057	−0.003	**0.042 ***	**0.009 ***
**Parity ^b^**								
Para 1	0.048	0.119	−0.057	−1.103	0.054	0.046	−0.429	−0.112
Multiparity	0.051	−0.497	−0.225	−0.356	−0.144	−0.229	**−0.927 ***	−0.375
**Education of mothers ^c^**	0.023	**0.341 ***	0.181	**0.578 ****	0.178	0.397 *	**0.436 ***	**0.196 ***
**Area of residence ^d^** (urban vs rural)	−0.101	−0.391	−0.47	**−1.302 ****	−0.036	−0.499	−0.591	−0.206
**Source of drinking water ^e^**								
Lake/shallow well	0.003	0.182	**0.929 ****	0.11	0.475	**1.311 ****	0.205	−0.038
Borehole	0.028	−0.03	**0.742 ***	0.397	0.388	**1.026 ***	0.324	−0.145
**Previous breast-feeding ^f^**	−0.013	**−0.643 ****	−0.2	0.026	−0.404	−0.225	0.342	0.16
**Beef consumption frequency ^g^**	0.017	−0.189	−0.071	−0.103	−0.077	−0.016	−0.211	−0.017
**Goat consumption frequency ^h^**	0.031	0.189	0.048	−0.164	−0.01	−0.007	−0.172	0.052

Values shown were simple linear regression analyses coefficients. ^a^ All POP concentrations were log 10 transformed before analysis. ^b^ Para 0 is the reference category. ^c^ None and primary educational level are the reference category. ^d^ Urban is the reference category. ^e^ Tap water is the reference category. ^f^ No previous breast-feeding is the reference category. ^g,h^ Less than twice a week is the reference category. * *p* < 0.05; ** *p* < 0.001. Significant findings are printed in bold.

**Table 4 ijerph-20-05289-t004:** Results of linear regression analysis measuring associations between maternal serum POP concentrations on birth outcomes.

	Outcomes	Maternal Serum POP Concentrations ^a^		
POP		*n*	*β* (95% CI)	*p*-Value
HCB	Head circumference (cm)	492	0.005 (−0.008 to 0.018)	0.447
	Birth length (cm)	492	0.001 (−0.005 to 0.007)	0.824
	Birth weight (kg)	522	0.001 (−0.055 to 0.057)	0.975
	Gestational age (weeks)	492	−0.001 (−0.189 to 0.016)	0.871
	Ponderal index (kg/m^3^)	491	−0.00007 (−0.0002 to −0.00007)	0.32
*p,p*′-DDE	Head circumference (cm)	492	−0.018 (−0.074 to 0.039)	0.542
	Birth length (cm)	492	−0.005 (−0.034 to 0.023)	0.708
	Birth weight (kg)	522	−0.043 (−0.303 to 0.217)	0.744
	Gestational age (weeks)	492	0.087 (0.008 to 0.166)	0.031
	Ponderal index (kg/m^3^)	491	0.00029 (−0.00034 to 0.0009)	0.367
t-NC	Head circumference (cm)	492	−0.053 (−0.105 to −0.0015)	0.044
	Birth length (cm)	492	−0.014 (−0.040 to 0.01199)	0.288
	Birth weight (kg)	522	0.073 (−0.167 to 0.313)	0.548
	Gestational age (weeks)	492	−0.002 (−0.074 to 0.069)	0.949
	Ponderal index (kg/m^3^)	491	0.0000151 (−0.001 to 0.001)	0.96
*p,p*′-DDT	Head circumference (cm)	492	0.006 (−0.055 to 0.067)	0.848
	Birth length (cm)	492	−0.013 (−0.043 to 0.017)	0.401
	Birth weight (kg)	522	0.139 (−0.139 to 0.417)	0.325
	Gestational age (weeks)	492	0.110 (0.026 to 0.193)	0.01
	Ponderal index (kg/m^3^)	491	0.0003 (−0.0004 to 0.0009)	0.45
Oxychlordane	Head circumference (cm)	492	−0.071 (−0.123 to −0.017)	0.01
	Birth length (cm)	492	−0.0187 (−0.045 to 0.007)	0.159
	Birth weight (kg)	522	0.1288 (−0.119 to 0.376)	0.307
	Gestational age (weeks)	492	0.016 (−0.059 to 0.092)	0.668
	Ponderal index (kg/m^3^)	491	0.0002 (−0.0004 to 0.0008)	0.575
cis-NC	Head circumference (cm)	492	−0.070 (−0.140 to −0.006)	0.048
	Birth length (cm)	492	−0.029 (−0.063 to 0.005)	0.108
	Birth weight (kg)	522	0.150 (−0.172 to 0.472)	0.36
	Gestational age (weeks)	492	0.033 (−0.066 to 0.131)	0.515
	Ponderal index (kg/m^3^)	491	0.0002 (−0.001 to 0.001)	0.578
*o,p*′-DDT	Head circumference (cm)	492	0.049 (−0.022 to 0.120)	0.173
	Birth length (cm)	492	−0.016 (−0.051 to 0.0187)	0.363
	Birth weight (kg)	522	0.199 (−0.123 to 0.521)	0.225
	Gestational age (weeks)	492	0.115 (0.016 to 0.213)	0.022
	Ponderal index (kg/m^3^)	491	−0.0001 (−0.001 to 0.001)	0.723
*o,p*′-DDE	Head circumference (cm)	492	−0.009 (−0.046 to 0.028)	0.617
	Birth length (cm)	492	−0.006 (−0.0238 to 0.012)	0.572
	Birth weight (kg)	522	0.127 (−0.040 to 0.293)	0.136
	Gestational age (weeks)	492	−0.004 (−0.047 to 0.055)	0.88
	Ponderal index (kg/m^3^)	491	0.000001 (−0.0004 to 0.0004)	0.995

**^a^** Maternal serum POP concentrations were log 10-transformed. All association between birth outcomes and POPs were adjusted for: maternal age, area of residence (urban vs. rural), maternal educational level, parity and source of drinking water.

## Data Availability

Data will be made available upon reasonable request from the corresponding author.

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
