# Peer review of "Predictors of Maternal Serum Concentrations for Selected Persistent Organic Pollutants (POPs) in Pregnant Women and Associations with Birth Outcomes: A Cross-Sectional Study from Southern Malawi"

_ijerph, 2023, doi:10.3390/ijerph20075289_

Round 1
Reviewer 1 Report
This manuscript describes a cross-sectional study of pregnant women to investigate predictors of maternal serum levels of Persistent organic pollutants (POPs) and the associations between POP levels and birth outcomes. The authors report multiple associations between serum levels and both maternal characteristics and birth outcomes.
This is a companion paper to a previous one already published (Int. J. Environ. Res. Public Health 2023, 20(3), 1689; https://doi.org/10.3390/ijerph20031689) and since this appears to be exactly the same population, this material probably should have been included in the previous paper rather than splitting the data up across several papers. The published paper is cited repeatedly within the methods section as providing further details on study design and methodology. Unfortunately, this is incorrect in several places. For example, the published paper does not provide any detailed information on the eligibility and exclusion criteria. 64 women were excluded from the study (Figure 1) and it is important to know why they were excluded and so these criteria need to be added to this manuscript. It is also unclear how many pregnant women were seen between August 2020 and July 2021 by the 3 hospitals mentioned. Is it 771 as mentioned in Figure 1 or is this the number actually assessed for eligibility? No information is provided on the LOD or LOQ of the assay methods in this manuscript. This is required since the analysis is based upon those POPs which were detected in >50% of the samples.
There are several additional issues that the authors need to address as follows
1. Please define all the abbreviations used.
2. Table 1 repeats data already present in their published paper and at best should be presented as supplementary information and not a full table in this manuscript.
3. Table 2: it is unclear whether the data presented is that for the samples that had detectable POP levels or from all the samples. Please clarify.
4. Table 3 is poorly presented with multiple rows of the same variable for different POPs. It would have been better presented as one single set of rows for the maternal characteristics and a column for each individual POP. This would enable much easier comparison between the results for different POPs which at present is extremely difficult to carry out.
5. Table 3 also completely lacks any data on the univariate analysis of the maternal characteristics and also misses out what could be key variables such as diet. There is no information as to why these variables were included in the multivariable analysis. Basically it appears that all the variables in Table 3 have been included in the model even if they weren’t significant. Why? The authors need to clearly justify their statistical approach.
6. Table 4 does not fit in with this paper. The authors must include an analysis of blood serum POP levels in relation to the diet of the pregnant women and not an indirect analysis linking diet to maternal educational levels.
7. The authors state lines 323-325 that “This information can be used to avoid adverse reproductive health outcomes as the greatest risk of exposure lies in this stage of development”. How? Do the authors suggest that urban dwellers must move to rural levels? Risk associated with lake/shallow well as a source of drinking water are increased for cis-NC but decreased for t-NC (Table 3). Are these results consistent and how can they be used to avoid adverse health outcomes?
Author Response
General comment:
This manuscript describes a cross-sectional study of pregnant women to investigate predictors of maternal serum levels of Persistent organic pollutants (POPs) and the associations between POP levels and birth outcomes. The authors report multiple associations between serum levels and both maternal characteristics and birth outcomes.
This is a companion paper to a previous one already published (Int. J. Environ. Res. Public Health 2023, 20(3), 1689; https://doi.org/10.3390/ijerph20031689) and since this appears to be exactly the same population, this material probably should have been included in the previous paper rather than splitting the data up across several papers. The published paper is cited repeatedly within the methods section as providing further details on study design and methodology. Unfortunately, this is incorrect in several places. For example, the published paper does not provide any detailed information on the eligibility and exclusion criteria. 64 women were excluded from the study (Figure 1) and it is important to know why they were excluded and so these criteria need to be added to this manuscript. It is also unclear how many pregnant women were seen between August 2020 and July 2021 by the 3 hospitals mentioned. Is it 771 as mentioned in Figure 1 or is this the number actually assessed for eligibility? No information is provided on the LOD or LOQ of the assay methods in this manuscript. This is required since the analysis is based upon those POPs which were detected in >50% of the samples.
General Response:We acknowledge that the analyses in the present manuscript is based on same study population from which we previously published on Poly- and Perfluoroalkyl substances (PFASs) in pregnant women and associations with birth outcomes. However, the authors decided not to include the present data as laboratory procedure were different hence the potential of making the manuscript cumbersome. This decision was arrived at based on the fact that sample preparation methods for the present analytes and PFASs weretotally different as also described and referred to in the method sections. Although, the instrumental analyses were both performed on a waters mass spectrometer, different chromatographical systems were used for separation together with different ionisation techniques and detection modes (gas chromatography with atmospheric pressure ionisation in positive mode for POPs vs liquid chromatography with electro-spray ionisation in negative mode for PFAS).
We have revised citation to our previous work accordingly and described the inclusion and exclusion criteria in detail. Furthermore, the figure 771 has been described to clear grey areas on recruitment process. Information on MDL has been added to the manuscript. see details on response number 3 below.
Responses to specific comments
- Please define all the abbreviations used.
Authors´ response: All abbreviations have now been defined. This included abbreviations for specific POPs (Lines 213 to 218).
- Table 1 repeats data already present in their published paper and at best should be presented as supplementary information and not a full table in this manuscript.
Authors´ response: Table 1 has now been removed from the manuscript. It will be uploaded separately as supplementary Table S2
- Table 2: it is unclear whether the data presented is that for the samples that had detectable POP levels or from all the samples. Please clarify.
Authors ´ response: Compounds with the DF over 50% were included for the statistical analysis. All the compounds with DF> 50% samples were included for the modelling, and the concentrations of selected compounds below the LOD were replaced by LOD/2. (Lines 169,172). In addition, Supplementary Table S1 which contains MDL concentrations for all POPs analysed has been provided.
- Table 3 is poorly presented with multiple rows of the same variable for different POPs. It would have been better presented as one single set of rows for the maternal characteristics and a column for each individual POP. This would enable much easier comparison between the results for different POPs which at present is extremely difficult to carry out.
Authors´ response: Table 3 has been revised. In this regard we now have table 2 (Univariable analysis) and 3 (Multivariable analysis). The table has been split as including data on the two analyses in one table resulted to a very big and difficult to read (Lines 252 to 268).
- Table 3 also completely lacks any data on the univariate analysis of the maternal characteristics and also misses out what could be key variables such as diet. There is no information as to why these variables were included in the multivariable analysis. Basically it appears that all the variables in Table 3 have been included in the model even if they weren’t significant. Why? The authors need to clearly justify their statistical approach.
Authors´ response: Diet has now been included in the both univariate and multivariable analyses (Lines 252 to 268).Furthermore, explanation on why other variables were either considered or not considered for multivariable analysis has also been added to the text(Lines 232 to 243).
- Table 4 does not fit in with this paper. The authors must include an analysis of blood serum POP levels in relation to the diet of the pregnant women and not an indirect analysis linking diet to maternal educational levels.
Authors´ response: Table 4 has been deleted completely.
- The authors state lines 323-325 that “This information can be used to avoid adverse reproductive health outcomes as the greatest risk of exposure lies in this stage of development”. How? Do the authors suggest that urban dwellers must move to rural levels? Risk associated with lake/shallow well as a source of drinking water are increased for cis-NC but decreased for t-NC (Table 3). Are these results consistent and how can they be used to avoid adverse health outcomes?
Authors´ response: The above sentence has been completely replaced with another (Lines 543 to 546).
Reviewer 2 Report
Lines 105-108. The authors refer to a previous publication for details pertaining to sample collection that are relevant to this manuscript. The reader should not have to read a previous publication in order to have all of the information pertinent to the current work. The citation as written here and in other places in the manuscript are not appropriate. It is entirely appropriate to cite the other work, but not in the context of “if you want the details, read our previous work”.
It is not clear what the pvalue in table 1 represents. Is this the difference between rural and urban? Is it just for the first line (i.e. gravidity (%), only significant difference is for gravidity of 1?).
In the methods section, multivariate analysis is described. In the paper, both multivariate and multivariable are discussed. It is unclear whether both types of analysis were performed and if so, this should be made clear, or if these terms are being used interchangeably, in which case, this should be corrected. Please see section 3.3 (text and title in table 3) and 3.5 for instances in which both methods are mentioned.
Table 3 would be more effective if the chemicals were listed across the top and the maternal characteristics along the side. This would allow direct comparison of the differences in chemical concentrations, it is hard to read organized as presented.
The statement that food is the primary source of chemical exposures should be referenced (line 202, 268). Additionally, the potential role of food consumption in the observed results (higher chemical concentrations in women with higher education) is strongly implied but there is no statistical analysis to determine if the food consumption is associated with higher levels of chemicals. If the data is available, this additional analysis should be included. This information would be more useful in regards to addressing the sources of exposure than level of education, and would suggest a potential implementable mitigation strategy to avoid certain foods rather than avoid higher education.
The potential role of access to these foods in the observed differences between rural and urban women is also not addressed.
Minor comments:
Sentence on line 174 is incomplete
It seems unnecessary to include the data in the text, see example below. It makes the sentence much harder to read and the information is available in the table.
participants educated up to either secondary or tertiary levels had significantly higher concentrations of p,p’-DDE (β = 0.364; 95% CI: 0.115 to 0.612; p = 0.004), 185 p,p’-DDT (β = 0.542; 95% CI: 0.274 to 0.809; p < 0.001), cis-NC (β = 0.340; 95% CI: 0.091 to 0.708; p=0.011), o,p’-DDT (β = 0.372; 95% CI: 0.061 to 0.683; p = 0.019) and o,p’-DDE (β = 0.200; 95% CI: 0.040 to 0.361; p = 0.014) than those who were only educated up to primary level.
Author Response
Reviewer 2
- Lines 105-108. The authors refer to a previous publication for details pertaining to sample collection that are relevant to this manuscript. The reader should not have to read a previous publication in order to have all of the information pertinent to the current work. The citation as written here and in other places in the manuscript are not appropriate. It is entirely appropriate to cite the other work, but not in the context of “if you want the details, read our previous work”.
Authors´ response:More detail has been added to the section on Maternal socio-demographic and neonates´ anthropogenic data. In this regard, the reader will now be able to know the above data without looking at our previous publication. The citation to our previous published work has also been revised (Lines 181 to 209). Just also to add that the table on Socio-demographic characteristics of recruited women and their neonateshas now been removed from the manuscript and will be attached separately as supplementally table S2 in accordance to comments from another reviewer.
- It is not clear what the pvalue in table 1 represents. Is this the difference between rural and urban? Is it just for the first line (i.e. gravidity (%), only significant difference is for gravidity of 1?).
Authors´ response: More description has been added to the footer of the table 1 (Footer for Supplementary Table S2).
- In the methods section, multivariate analysis is described. In the paper, both multivariate and multivariable are discussed. It is unclear whether both types of analysis were performed and if so, this should be made clear, or if these terms are being used interchangeably, in which case, this should be corrected. Please see section 3.3 (text and title in table 3) and 3.5 for instances in which both methods are mentioned.
Authors´ response: The researchers used multivariable linear regression analysis and NOTMultivariate analysis. In this regard, this has been corrected in the manuscript accordingly (Lines 175,177 and 310).
- Table 3 would be more effective if the chemicals were listed across the top and the maternal characteristics along the side. This would allow direct comparison of the differences in chemical concentrations, it is hard to read organized as presented.
Authors´ response: Table 3 has been revised accordingly. Just also to add we now have two tables for linear regression of POP concentrations in blood serum and maternal characteristics;table 2 (Univariable analysis) and 3 (Multivariable analysis). This was due to the fact that another reviewer also asked for univariable analysis data. The table has been split as including data on the two analyses in one table resulted to a very big and difficult to read (Lines 252 to 268).
- The statement that food is the primary source of chemical exposures should be referenced (line 202, 268). Additionally, the potential role of food consumption in the observed results (higher chemical concentrations in women with higher education) is strongly implied but there is no statistical analysis to determine if the food consumption is associated with higher levels of chemicals. If the data is available, this additional analysis should be included. This information would be more useful in regards to addressing the sources of exposure than level of education, and would suggest a potential implementable mitigation strategy to avoid certain foods rather than avoid higher education.
Authors´ response: References on food as the primary source POPs has been added (Lines 293 and 439).
The authors revisited the analysis. Diet has now been included in the both univariable and multivariable analyses (Lines 232 to 245). Furthermore, explanation on why other variables were either considered or not considered for multivariable analysis has also been added to the text. Multivariable analysis showed no significant association between specific food and POPs levels (Table 3, Lines 264 to 265) In this regard, claims on the effect of food on POPs concentrations in maternal serum have been removed from the manuscript.
- The potential role of access to these foods in the observed differences between rural and urban women is also not addressed.
Authors´ response: This has been addressed from line number 449 to 501.
Minor comments:
- Sentence on line 174 is incomplete
Authors´ response: The sentence has now been completed.
- It seems unnecessary to include the data in the text, see example below. It makes the sentence much harder to read and the information is available in the table.
participants educated up to either secondary or tertiary levels had significantly higher concentrations of p,p’-DDE (β = 0.364; 95% CI: 0.115 to 0.612; p = 0.004), 185 p,p’-DDT (β = 0.542; 95% CI: 0.274 to 0.809; p < 0.001), cis-NC (β = 0.340; 95% CI: 0.091 to 0.708; p=0.011), o,p’-DDT (β = 0.372; 95% CI: 0.061 to 0.683; p = 0.019) and o,p’-DDE (β = 0.200; 95% CI: 0.040 to 0.361; p = 0.014) than those who were only educated up to primary level.
Authors´ response: This has been revised. The authors have now just reported p-values in areas in which a lot of variables have been reported in a single sentence like in the example above. On the other have we would like to point out that Table 3 has been revised to tables 2 and 3, following comments from reviewers. These tables new tables only display coefficients and significance level and not the confidence intervals and p-values. In this regard, the authors were wondering if is still appropriate to change the way the estimates are being reported in the text hence not changed this in some parts of the text.
Reviewer 3 Report
Title: Predictors of Maternal Serum Concentrations for Selected Persistent Organic Pollutants (POPs) in Pregnant Women and Associations with Birth Outcomes: A Cross-Sectional Study from Southern Malawi
General comments
The manuscript addresses an interesting issue by exploring the relation between the levels of selected Persistent Organic pollutants in pregnant women from Malawi, with birth outcomes. The study provides an adequately descriptive profile of the situation in the country and therefore is very interesting for publishing in International Journal of Environmental Research and Public Health. Studies from countries of the Southern hemisphere is quite rare and they are very important concerning to the global health in Earth.
Statistical analysis has been performed using quite traditional tools (multivariate regression models) although as the datasets are really big covering whole country the paper is interesting.
The paper is well written and organized while the statistical tools have been well explained and discussed with the available literature. The topic is related with public health and the very interesting effects of some pollutants to the birth outcomes thus I think that is suitable for publication after revising some points.
In general, I would suggest authors to revise the whole manuscript as far as the English used. Some specific suggestions will be included in each section’s comments.
Abstract
Line 28 “Report on” Please modify the phrasal verb.
Results
Lines 212 “concentrations” in title
Discussion
The first paragraph is a repetition of the results section. I think that should be absorbed in the whole body of discussion section
Line 232 Authors could replace “were” with “Were found to be”
Line 244,253,265,288 “consistent” I think authors could use other words also for the comparison with literature
Line 276-277 and 294-295 The phrases should be rephrased
Line 280 “Statistically significantly” Please correct
Line 286 A possible explanation should be added by the authors
Conclusions
Line 317-319 Please rephrase
Author Response
Reviwer 3
- Abstract
Line 28 “Report on” Please modify the phrasal verb.
Authors´ response: This has been revised. The sentence now reads “We are reporting on …" (Lines 28 and 29).
- Results
Lines 212 “concentrations” in title
Authors´ response: This has been corrected (Line 306).
- Discussion
The first paragraph is a repetition of the results section. I think that should be absorbed in the whole body of discussion section
Authors´ response: This is acknowledged. However, the authors were just trying to follow the standard practice of summarizing the findings in the first paragraph of the discussion just to remind the reader on the main finding before going deeper in the detailed discussion of results.
Line 232 Authors could replace “were” with “Were found to be”
Authors´ response: Corrected in the manuscript (Line 405).
Line 244,253,265,288 “consistent” I think authors could use other words also for the comparison with literature
Authors´ response: Other phrases like; “Thesefindings are similar to” and “These results are similar to...” have been used (Lines 424 and 503).
Line 276-277 and 294-295 The phrases should be rephrased
Authors´ response: A number of sentences in this paragraph have been restructured to harmonize the flow (Lines 449 to 501).
Line 280 “Statistically significantly” Please correct
Authors´ response: Corrected.
Line 286 A possible explanation should be added by the authors
Authors´ response: Possible explanation has now been added.
- Conclusions
Line 317-319 Please rephrase
Authors´ response: This has been rephrased. It now reads “This study gives a foundation upon which the knowledge on main predictors for POPs levels in maternal serum among pregnant women at delivery stage from low-middle income and African settings can be built on(Lines 537 to 540).
Reviewer 4 Report
In this study, Mwapasa et al. investigated the associations between maternal POP concentrations in serum and birth outcomes. The study design is straightforward and the results revealed a number of associations between POPs and birth outcomes in Malawi. However, conflict data do exist. My comments/suggestions regarding this manuscript are detailed as follows:
l In the abstract, the authors indicated that the multiparity was positively associated with POP concentrations. However, the authors showed that the parity is negatively associated with maternal serum concentrations of POPs. Please clarify.
* Please include relevant discussion about the observed associations among POP concentrations, gestational age, and head circumference. Is there any mechanistic evidence that can support the observed associations?
Author Response
Reviewer 4
- In the abstract, the authors indicated that the multiparity was positively associated with POP concentrations. However, the authors showed that the parity is negatively associated with maternal serum concentrations of POPs. Please clarify.
Authors´ response: The typographic error in the abstract has been corrected to tally with what is in the was written in the results section. The correct statement is that parity was negatively associated with POP concentrations (Line 34).
- * Please include relevant discussion about the observed associations among POP concentrations, gestational age, and head circumference. Is there any mechanisticevidence that can support the observed associations?
Authors´ response: Explanation on possible reason to the observed association has been added (514 to 521 and 523 to 526).
Round 2
Reviewer 1 Report
The authors need to
1. Add as a footnote to Table 1 that the missing data has been inputted as LOD/2
2. Justify the use of LOD/2 as the value for the missing data rather than other methods that could be used.
Author Response
- English language and style are fine/minor spell check required.
Authors´ response:Minor spell check was executed (Lines 151, 174, 177, 178,183 and 220).
- Add as a footnote to Table 1 that the missing data has been inputted as LOD/2.
Authors´ response: Footnote added to Table 1 (Line 206).
- Justify the use of LOD/2 as the value for the missing data rather than other methods that could be used.
Authors´ response: Justification now added to the manuscript (Line 148 to 150).
Reviewer 2 Report
Authors responses to previous comments are acceptable.
Author Response
- English language and style are fine/minor spell check required.
Authors´ response:Minor spell check was executed (Lines 151, 174, 177, 178,183 and 220).
Reviewer 4 Report
The authors did a good job in revising the manuscript. I do not have further questions.
Author Response

(The authors gave the same response as above.)
